# Optical Sensing of Humidity Using Polymer Top-Covered Bragg Stacks and Polymer/Metal Thin Film Structures

**DOI:** 10.3390/nano9060875

**Published:** 2019-06-10

**Authors:** Katerina Lazarova, Darinka Christova, Rosen Georgiev, Biliana Georgieva, Tsvetanka Babeva

**Affiliations:** 1Institute of Optical Materials and Technologies ‘‘Acad. J. Malinowski’’, Bulgarian Academy of Sciences, Akad. G. Bonchev str., bl. 109, 1113 Sofia, Bulgaria; rgeorgiev@iomt.bas.bg (R.G.); biliana@iomt.bas.bg (B.G.); 2Institute of Polymers, Bulgarian Academy of Sciences, Akad. G. Bonchev Str., bl. 103-A, 1113 Sofia, Bulgaria; dchristo@polymer.bas.bg

**Keywords:** optical sensing, humidity, Bragg stacks, branched polymers

## Abstract

Thin films with nanometer thicknesses in the range 100–400 nm are prepared from double hydrophilic copolymers of complex branched structures containing poly(*N*,*N*-dimethyl acrylamide) and poly(ethylene oxide) blocks and are used as humidity sensitive media. Instead of using glass or opaque wafer for substrates, polymer thin films are deposited on Bragg stacks and thin (30 nm) sputtered Au–Pd films thus bringing color for the colorless polymer/glass system and enabling transmittance measurements for humidity sensing. All samples are characterized by transmittance measurements at different humidity levels in the range from 5% to 90% relative humidity. Additionally, the humidity induced color change is studied by calculating the color coordinates at different relative humidity using measured spectra of transmittance or reflectance. A special attention is paid to the selection of wavelength(s) of measurements and discriminating between different humidity levels when sensing is performed by measuring transmittance at fixed wavelengths. The influence of initial film thickness, sensor architecture, and measuring configuration on sensitivity is studied. The potential and advantages of using top covered Bragg stacks and polymer/metal thin film structures as humidity sensors with simple optical read-outs are demonstrated and discussed.

## 1. Introduction

Humidity is present everywhere and is one of the mostly important physical parameters in our lives. The accurate monitoring of humidity is significant in semiconductors, electronics, food processing, and pharmaceutical industries where the quality of products is influenced sufficiently by the humidity. Humidity control is also essential in museums or archives in order to ensure the correct storage of artworks. Further, monitoring of relative humidity in office environments is in favor of human comfort and health and helps achieve hygienic conditions.

The perfect humidity sensor should have high sensitivity, fast response, long-term durability, low cost, easy detection, and wide dynamic range, i.e., it should operate over a wide range of humidity. The established technology nowadays is an electrical measurement of humidity [1] where the change of resistivity, dielectric constant, or the impedance of a sensitive medium, usually metal oxide thin film, is used [2,3].

In recent years, optical sensing of humidity has gained increasing interest and great research efforts have been put in this field [4,5]. Optical sensors operate at room temperature and do not require an electrical power supply, which makes them suitable for flammable and harsh environments and enables operation at high pressures as well. Optical fiber humidity sensors are the most widely spread optical sensors for humidity. Different materials have already been used for their functionalization such as agarose gel [6], graphene oxide [7], polyvinyl alcohol [8], etc. The main disadvantage of this type of sensor is the cost of optical equipment, particularly spectrometers and optical spectrum analyzers.

Cost effective and relatively simpler approaches are color sensing of humidity or monitoring of appropriately chosen optical parameters that change with humidity. In the first approach the detection is performed by visual inspection of color, or color monitoring by an inexpensive camera [9,10,11]. Different photonic structures have been utilized: reflection holograms [9,10], Bragg stacks [12], and 2D and 3D photonic crystals [11,13]. In the second approach the swelling of a sensitive medium [14] or the transmittance power [15] are measured with humidity.

Polymers attract considerable interest as optical sensing materials due to the advantageous features such as flexibility, light weight, ease of processability, useful mechanical properties, and compatibility with oxides and ceramics [2,3]. In general, natural or synthetic polymers such as hydroxyethyl cellulose, chitosan, poly(methyl methacrylate), polyaniline, poly(vinyl alcohol), and poly(ethylene oxide) are used in optical sensors in the form of films, beads, micro-rods, nano-spheres, etc. [16,17,18]. Along with the use of single commodity polymers and polymer–inorganic composites, various copolymers [19,20,21,22], hybrid structures [23,24], and smart copolymers [25] are synthesized and their optical properties studied in an attempt to address the specific requirements arising from the diverse potential applications.

It is already demonstrated that thin films with submicron thicknesses exhibit faster swelling than bulk hydrogels [26]. We have already shown that in order to achieve a large response in terms of swelling, the polymer thin film must have the right chemistry and the right macromolecular structure [22]. However, the geometry of measurements and the design of the sensor, for example thickness, substrate type, sublayer, top layer, etc., also have considerable impact on sensor sensitivity.

In this paper, for detection of humidity we use transmittance measurements because they are simpler, more accurate, and less expensive compared to reflectance measurements. Moreover, observing sample color in transmittance mode, i.e., using the light transmitted through the sample, could overcome to a great extent the ambiguity related to the angle dependence of the color when reflectance mode is used. We study two sensor configurations. In the first one we deposit the sensitive film on top of the Bragg stack, instead of incorporating it in the stack’s backbone, thus overcoming possible issues of incompatibility that may arise when organic and inorganic materials are used for the stack. The second sensor configuration is comprised of a thin metal sublayer deposited on glass substrate and top-covered with the sensitive polymer film. The metal sublayer has dual purposes: to provide color for colorless polymer film and to increase the optical contrast between the polymer and glass substrate that have almost the same refractive indices (around 1.5).

In this study we prove the concept of utilization of top covered Bragg stacks and polymer/metal thin films as optical sensors for humidity. As humidity sensitive media we use thin films of previously developed multiblock copolymer with branched macromolecular architecture comprised of poly(*N*,*N*-dimethyl acrylamide) (PDMA) and poly(ethylene oxide) (PEO) segments [22]. The humidity sensing ability are demonstrated through both transmittance measurements and calculation of CIE (International Commission on Illumination) color coordinates at different relative humidity in the range 5–90% relative humidity (RH). The impact of initial film thickness, sensor architecture, and measuring configuration on sensitivity is studied and discussed.

## 2. Materials and Methods

The monomer *N*,*N*-dimethylacrylamide (DMA) was purchased from Sigma–Aldrich (Steinheim, Germany) and purified from inhibitor by passing through a column of basic alumina before use. Poly(ethylene oxide) of molar mass 2000 g·mol^–1^ (PEO2000) was supplied from Fluka (Buchs, Switzerland). Initiator diammonium cerium (IV) nitrate (NH_4_)_2_Ce(NO_3_)_6_ and crosslinker poly(ethylene glycol) diacrylate of reported average Mn 575 (PEGDA) were purchased from Sigma–Aldrich as well. Other solvents and reagents were of standard laboratory reagent grade and used as received.

The synthesis of the copolymer is described in detail in our previous papers [22,27]. Briefly, a redox polymerization of *N,N*-dimethylacrylamide (DMA) in deionized water was carried out using ammonium cerium (IV) nitrate as the initiator. Poly(ethylene oxide) and poly(ethylene glycol) diacrylate were implemented as a hydroxyl functionalized initiating moiety and as a cross-linker, respectively. The polymerization was carried out for 3 hours at 35 °C in nitrogen atmosphere under vigorous stirring and terminated by diluting the reaction mixture with methanol (1:1 volume ratio). The copolymer solution thus obtained was used for thin polymer film deposition without further isolation or purification. In order to control the thickness of the films, the copolymer solution was diluted to the appropriate concentration by using a water/methanol mixture (1:1 volume ratio). Chemical structure of the copolymer is schematically presented in Appendix A.

Thin polymer films with thickness of 100, 200, 300, and 400 nm were deposited by the spin-coating method at a rotation speed of 4000 s^−1^ and duration of 60 s using polymer solutions with concentrations of 4, 6.25, 7.3, and 8.3 g L^−1^, respectively. A post deposition annealing at 180 °C for 30 min in air was applied to all films. Silicon wafers, Bragg reflectors, and Au–Pd coated glasses were used as substrates. Five- and seven-layer Bragg reflectors were prepared on glass substrates by alternating deposition of sol-gel Nb_2_O_5_ [28] and SiO_2_ [29] films or dense and porous Nb_2_O_5_ films [30]. The reflectors were designed in a way that their stop bands were in the visible spectral range. The Au–Pd sublayers with thicknesses of 15, 30, and 70 nm were deposited on glass substrates by cathode sputtering of gold/palladium target (Quorum Technologies) for 15, 60, and 90 seconds, respectively, under vacuum 4 × 10^−2^ mbar using a Mini Sputter Coater SC7620 system. The thickness and diameter of the target were 0.2 and 57 mm, respectively. The purities of gold and palladium were 99.99% and 100%, respectively, and Au:Pd ratio was 80:20. The Au–Pd layers’ thickness was determined using mechanical profilometer Talystep (Rank Taylor Hobson, Leicester, UK) and confirmed by 3D optical profiler (Zeta 20, Zeta Instruments, Milpitas, CA, USA).

Refractive index, extinction coefficient, and thickness of polymer films were calculated using previously developed two-stage nonlinear curve fitting of reflectance spectra measured with UV-VIS-NIR spectrophotometer (Cary 5E, Varian, Melbourne, Australia) at normal light incidence [28]. The accuracy is 0.01, 0.005, and 1 nm for the refractive index, extinction coefficient, and thickness, respectively. The sensing behavior was tested by measuring in-situ transmittance or reflectance spectra at different levels of relative humidity (RH), realized using a homemade bubbler system that generates vapors from liquids [31]. The exact values of RH were obtained from a reference humidity sensor placed in the measuring cell. In order to check the applicability of the studied samples for color sensing of humidity, CIE color coordinates were calculated at each humidity level [32] and plotted on X–Y color space. Measured transmittance or reflectance spectra of the samples were used for calculation of CIE color coordinates.

Indeed, the reflectance and transmittance spectra of the obtained thin polymer films were registered in at least three consecutive experiments for each film sample. As the reflectance and transmittance data points derived from the subsequent experiments were practically equal, standard deviations were not added to the figures in the manuscript. The reported accuracy of the spectrophotometer measurement is 0.1% for transmittance and 0.26% for reflectance.

The sensitivity of the sensors *S*, measured in % per % RH, was calculated according to the following equation:(1)$$S\;=\;\frac{\Delta R}{{\mathrm{RH}}_{2}-{\mathrm{RH}}_{1}}$$
where Δ*R* (or Δ*T*, if transmittance *T* is measured) is the change of sensor’s reflectance (or transmittance) in % which is provoked by the alteration of relative humidity from RH_1_ to RH_2_.

The accuracy/resolution of detection (Δ*RH*), measured in % RH, was calculated by Equation (2):(2)$$\Delta RH\;=\;\frac{errR\;\left(\%\right)}{S\;\left(\%/\%RH\right)}$$
where *errR* (or *errT*, if *T* is measured) is the experimental error (measurement accuracy) of *R* or *T* and *S* is the sensitivity, calculated by Equation (1).

## 3. Results and Discussion

When polymer film or hydrogel are exposed to high humidity the polymer chains swell due to the penetration of moisture inside the film and as a result they change their thickness, i.e., swelling is observed. Simultaneously, a drop in refractive index takes place due to the reduced density of the film [32]. It may be expected that the degree of swelling depends on the initial thickness of the film. Therefore, the first step of our investigation concerns this dependence. Two possible approaches for conducting this study exist: measuring reflectance or transmittance of the films at different humidity levels or to calculate the change of film thickness with humidity. In our work we choose to monitor the thickness change because film thickness is an intrinsic physical parameter and, unlike transmittance and reflectance, it is independent of other parameters such as refractive index, wavelength, substrate type, etc.

Thin polymer films with different thicknesses (from 100 to 400 nm) were deposited on Si-substrate and exposed sequentially to relative humidity of 5% RH and 70% RH. Reflectance spectra of the films were measured at both humidity levels and optical constants and film thicknesses were determined using non-linear curve fitting [28]. Reflectance spectra of 100 nm and 300 nm thick films are presented in Figure 1a,b, respectively. The increase of the number of interference peaks and decrease of their intensity are clearly seen in the case of thicker films (Figure 1b). The first observation indicates the increase of the thickness of the film when it is exposed to high humidity levels, while the decrease of fringe intensity confirms the decrease of the refractive index associated with the drop of film density that is in correlation with our previous results [32]. The relative thickness increase Δ*d* (in percent), i.e., the swelling, is calculated according to Equation (3):$$\Delta d=\left(\frac{d_{70\%}-d_{5\%}}{d_{5\%}}\right)\times 100$$
where *d*_70%_ and *d*_5%_ are the calculated film thicknesses at relative humidity of 70% RH and 5% RH, respectively.

Figure 1c presents relative thickness change as a function of initial polymer film thickness, i.e., the film thickness at 5% RH. It is seen that for all samples the exposure from low to high humidity leads to an increase of the films thickness. However, the degree of swelling depends strongly on the initial thickness of the films and it is the highest for 300 nm thick film. It is seen that Δ*d* increases with thickness: 26%, 73%, and 97% for polymer films with initial thicknesses of 100, 200, and 300 nm, respectively. For the film with initial thickness of 400 nm a small decrease in Δ*d* is observed (84%) compared to those of an initial thickness of 300 nm. The possible reason for the thickness dependence of the swelling is the adhesion to the substrate. It could be expected that the influence of substrate adhesion is the strongest for the thinnest film and decreases gradually with thickness. This explains the fourfold increase in swelling of polymer film with a thickness of 300 nm, as compared to this with a thickness of 100 nm.

In order to determine the response time of both thin (100 nm) and thick (300 nm) polymer films we deposited them on glass substrate and monitored the temporal change of transmittance at fixed wavelength (λ_max_) when the humidity in the measuring cell goes continuously from 5% RH to 80% RH. Figure 2a presents the temporal change of the humidity in the cell measured by the reference humidity sensor. Figure 2b presents the temporal dynamic of transmittance of thin polymer films with thicknesses of 100 and 300 nm deposited on glass substrate measured at selected wavelengths when the humidity in the cell changed continuously from 5% RH to 80% RH following the dependence presented in Figure 2a. The monitoring wavelength λ_max_ is selected as the wavelength at which the humidity response is the strongest. Considering that the transmittance and reflectance of thin films are non-linear functions of film thickness (Appendix A) it is obvious that λ_max_ depends on the film’s optical constants and thickness as well as on the type of substrate used. Glass substrate is selected for this experiment because the oscillatory behavior of the transmittance versus thickness curve is not so strong as compared to the case of the reflectance versus thickness curve when silicon substrate is used (Appendix A).

To study the delay of our sensors relative to the reference sensor, we normalized the measured humidity and transmittance curves and plotted them in Figure 2c. The comparison to a reference sensor shows a delay for low humidity, i.e., at the beginning of measurements that vanishes for RH values greater than 50% RH. For example at 10% RH and 20% RH the delay is 8 and 6 s, respectively, and it decreases to 4 s for 30% RH and 40% RH. Interestingly both polymer films react similarly to the change of humidity and there is no delay for the thicker film in respect to the thinner one. For a rapid change of humidity from 50% RH to 95% RH, achieved by blowing of humid air, the change of color is almost instant. The sensor recovers rapidly when blowing is off.

We have already demonstrated that when humidity is increased, the thickness of all studied polymer films increases as compared to its initial value. Besides, the swelling depends on the initial film thickness and has a maximal value for 300 nm thick film. Further, it is interesting to study how the film thickness changes with humidity, i.e., the humidity dependence of film swelling.

Reflectance spectra at relative humidity of 5, 40, 60, and 90% RH of thin polymer films with thickness of 100, 200, and 300 nm deposited on silicon substrate are presented in Figure 3a–c. For all films substantial change of reflectance spectra with RH is observed. Spectra are further used for calculation of the thickness of the films [28].

The swelling dynamic, i.e., the growth of film thickness with relative humidity, is shown in Figure 3d–f. For the thinnest film an almost linear dependence of *d* versus RH is obtained in the whole studied RH range (5–90% RH). This is beneficial for using the film as a sensing medium for a wide range of humidity. Similar linear dependence is obtained for 200 nm thick film in 40–90% RH. The expansion of the thickest film is exponential with the increase of the relative humidity and changes from 314 nm at 5% RH to 965 nm at 90% RH. The overall change of film thicknesses is 31%, 127%, and 207% for 100, 200, and 300 nm thick films, respectively.

Considering the substantial change in thickness with humidity, we could expect also considerable change of color. If it turns out this is the case, the films could be used as color indicators for humidity. However, first the evolution of color with humidity should be investigated. Using the measured reflectance spectra of films at different humidity we calculated the CIE color coordinates [32] and plotted them in X–Y color space (Figure 4).

The color of the thinnest film is almost the same for RH values in the range from 5% to 40% RH and starts to change gradually from dark blue to light blue for RH higher than 40%. The color coordinates are well separated in the color space thus enabling color sensing of humidity in the range 40–90% RH. Although the thickness values at 5% RH and 90% RH are quite different for thicker films, 202 and 456 nm, respectively, the colors of the films at these humidity levels are very similar (Figure 4b—red and green circles are close to each other). The reason is the periodicity of transmittance and reflectance and respectively of colors with optical thickness of the films (see Appendix A). Remaining points are well separated in X–Y color space (Figure 4b). This means that distinct colors will be observed in a wide humidity range, thus making the 200 nm film suitable for color sensing. When analyzing the color coordinates with humidity for the thickest film (300 nm) (Figure 4c) a conclusion could be made that it is suitable for color sensing in the range 5–60% RH. The sensitivity will be the highest because the separation of the points is the most distinguished and unambiguous in this case.

The main drawback when color sensing is used is the dependence of the color on the viewing angle. The observer cannot be sure whether the color change is due to the humidity change or it is because of his position during the color inspection. It is well known that reflectance and transmittance are functions of the incident angle of the light. Because the color perception of the observer is associated with reflectance or transmittance spectra, their change with incident angle of light results in different colors observed at different viewing angles. For example Figure 5 presents pictures of polymer films with different thicknesses deposited on silicon substrate at three viewing angles—10, 30, and 40 degrees. The change of the color with viewing angle for the thickest film is well seen. Fortunately, for thinner films the color changes are not so distinctive. Indeed, for the thinnest one the color change with angle is slightly noticeable.

Another possible approach to sensing is monitoring of signal (transmittance, *T* or reflectance, *R*) at a fixed wavelength, λ_max_. As mentioned above, because *T* and *R* are nonlinear functions of *n*, *k*, *d,* and λ, the selection of λ_max_ is not a trivial task and has an important role in detection. For example, if we pick a wavelength of 550 nm for λ_max_ for polymer with a thickness of 100 nm (Figure 3a), the measured *R* will increase gradually with RH in an approximately linear manner. However, if λ_max_ = 700 nm than *R* versus RH dependence will exhibit strong nonlinear behavior.

The humidity dependence of reflectance for different λ_max_ is shown in Figure 6 for all studied samples. For the thinnest one (Figure 6a), the most appropriate λ_max_ is 550 nm. It is seen that at a wavelength of 550 nm the *R* versus RH dependence is almost linear in the whole humidity range (Figure 6d) and the calculated sensitivity is 0.13% (Equation (1)). If one assumes that the reflectance measurement error is 0.26%, then the lowest humidity step that could be discriminated by the sensor will be 2% RH. Further, if 300 nm thick polymer film is used as sensitive medium and the reflectance of the film is monitored at λ_max_ of 600 nm a twice as high sensitivity of 0.26% could be achieved (the higher incline of the curve *R* vs. RH in this case is well seen in Figure 6а). This means that the resolution of the sensor will be 1% RH, which is a very good achievement. Unfortunately, the dynamic range of the sensor is limited. This film can be utilized for humidity measurements up to 50% RH.

From a technological point of view it is more convenient to monitor the transmittance of the film rather than the reflectance. Transmittance measurements are easier to be performed. Besides, they are more accurate and inexpensive. When signal is measured in transmittance mode the angle of incidence could be zero (the so called normal incidence) and the ambiguities associated with angle dependence of color are overcome to the great extent. However, to make transmittance measurements possible, a transparent substrate is required. The cheapest option is to use glass or plastic substrates but in our case they are not suitable because their refractive indexes match the refractive index of polymer thin film and the accuracy drops substantially. Besides, no color will be observed.

Recently we have shown [27,32] that Bragg stacks that exhibit structural colors could be used as transparent substrates. Briefly, Bragg stacks are multilayered systems comprised of alternating layers with low and high refractive indices and quarter-wavelength optical thickness. Because of the quarter-wavelength optical thickness all waves reflected from the layer boundaries in the stack are in phase and interfere constructively. As a result a band of high reflectance appears that is responsible for the observed coloring of the stack. On the contrary, all waves transmitted from different boundaries are out of phase and a band of low transmittance, called a stop band, is generated.

Another advantage of using a Bragg stack as a transparent substrate is illustrated in Figure 7a which presents the pictures of 5- and 7-layer Bragg stacks covered with 300 nm thick polymer films at three different viewing angles. It is seen that the change of the color with viewing angle is weaker as compared to the case of the same film deposited on silicon substrate (Figure 5). This will be beneficial for utilization of polymer top-covered Bragg stacks for color sensing of humidity.

In order to achieve the highest sensitivity, we optimized the polymer film thickness through theoretical modeling. The calculated humidity induced change in transmittance (Δ*T*) of 5-layer Bragg stacks covered with polymer films with different thicknesses in the range 100–400 nm is shown in Figure 7b. It is seen that Δ*T* increases with the thickness of the polymer film reaching a steady state for thicknesses higher than 250 nm. Therefore, in the next step of our investigation we use polymer films with a thickness of 300 nm and deposit them on four different Bragg stacks comprised of five and seven alternating layers of Nb_2_O_5_/SiO_2_ and dense/mesoporous Nb_2_O_5_. Our previous results have indicated that the characteristics of stacks, such as operating wavelength, number of layers, optical contrast, etc., do not substantially influence the sensitivity [27]. Still, the best results were obtained for the 5-layer stack consisting of Nb_2_O_5_ and SiO_2_ layers. 

Transmittance spectra of the sample at different humidity levels are presented in Figure 8a. In the RH range 5–25% the spectra are almost the same. With increasing humidity, a substantial change is observed. For RH = 30% the shape of the spectra changes; the long-wavelength peak almost disappears, while the short-wavelength one becomes more distinct. With further increase of humidity a shift of the peak toward longer wavelengths takes place.

The detailed study of the spectra reveals that the most appropriate wavelengths (λ_max_) for measuring *T* are 650 and 730 nm (Figure 8b). The obtained sensitivities are 0.37% and 0.13% with dynamic ranges of 40–80% RH and 25–70% RH, respectively. Very good linear dependence of *T* versus RH is obtained in both cases. Furthermore, the achieved resolutions of detection are very high: 0.3% RH in the 40–80% RH range and 0.8% RH at 25–70% RH. They are calculated with the aid of Equation (2), considering 0.1% experimental error in *T*.

Although the sensitivity and resolution are high, an ambiguity exists due to the same *T*-values at different RH (dotted horizontal lines in Figure 8b). For example, the same *T* will be measured at 25% RH and 54% RH, or at 64% RH and 90% RH, or at 70% RH and 84% RH (the dotted horizontal lines) and the observer will need additional measurements to discriminate between them. We have shown that this could be overcome if color sensing is used simultaneously. Figure 8c presents the calculated color coordinates in transmission mode for relative humidity of 25% and 54%. The points are well separated in the color space thus assuring unambiguous detection of humidity because the observed color of the sample will be different at 25% and 54% and it will not be difficult to discriminate between the two RH levels. The results for other points of ambiguity are very similar and are presented in Appendix A.

Another approach that we consider for humidity sensing by transmittance measurements is implementation of thin metal film deposited on transparent substrate as a transducer. The metal film should be sufficiently thin in order to allow transmittance measurements to be performed. Figure 9a presents transmittance spectra of polymer/metal structures with varying thickness of Au–Pd thin films (15, 30, and 70 nm) and polymer film with a thickness of 300 nm deposited on top. It is seen that *T* decreases with the thickness of the Au–Pd film from 74% to 33% (the values are at a wavelength of 650 nm). For further experiments we selected Au–Pd film with a thickness of 30 nm because it has an average transmittance of 50%. The change of *T* spectra with humidity is shown in Figure 9b. As in the case of *R* of single film on opaque substrate (Figure 3a–c), the number of the interference peaks increases and their intensity decreases, which indicates an increase of thickness and a decrease of refractive index with humidity.

The dependence of *T* on humidity for the polymer/metal system is shown in Figure 10a. The most appropriate wavelength for measuring *T* is 700 nm. Two linear ranges in *T* versus RH dependence are clearly seen. The sensitivity is 0.14% for RH = 25–70% and increases to 0.39% for higher humidity (RH = 70–90%). The respective resolutions are 0.7% RH and 0.3% RH. As in the case above, when using samples comprised of polymer and metal film, ambiguity in the sensing also exists due to the same *T*-values at different RH. In this case, the discrimination between different RH values is also possible if the color of the sample is monitored simultaneously. Figure 10b,c illustrates the different colors, observed in transmission mode, for the two pairs of RH: (25%, 87%) and (40%, 81%), respectively.

Instead of observation of color, the discrimination between RH with the same transmittance values is possible if a second wavelength is used for monitoring of *T*. As an example, the values of *T* measured at wavelength of 430 nm are presented as blue numbers in Figure 10a. It is seen that the difference varies between 2.2% and 4.4%, which is sufficient to distinguish different RH exhibiting the same *T* values at 700 nm.

The obtained results are summarized in Table 1. The widest dynamic range of sensing is obtained for polymer with a thickness of 100 nm deposited on silicon substrate. For sensing, monitoring of reflectance should be applied. Further, it is seen that the implementation of thin metal film as a substrate for 300 nm polymer film is beneficial for increasing the sensitivity and resolution of humidity sensing and enables monitoring of transmittance instead of reflectance. The narrowing of dynamic range is acceptable.

As mentioned in the Introduction section, the most widely spread and sensitive technology for humidity sensing is fiber optic sensing. Unfortunately they have disadvantages of relatively high cost of optical equipment, particularly spectrometers and optical spectrum analyzers, and complicated preparation of sensitive elements, especially in the case of side-polished optical fibers. The sensitivity of fiber optic sensors is calculated as nm per % RH because usually for detection the wavelength shift of the signal dip is monitored. The sensitivity values reported in the literature vary in a broad range from 0.023 nm/% RH to 1.01 nm/% RH [33]. In order to make a comparison with our sensor, we calculated the resolution using Equation (2) assuming wavelength accuracy of 0.1 nm. The calculated values vary from 4.3% RH to 0.1% RH. Therefore, the value of 0.3% RH achieved by our sensor is very close to the best reported resolution of optical humidity sensing so far. Furthermore, the proposed sensor has the advantages of simple preparation and simple detection. The good sensitivity and resolution along with the technological convenience make the presented sensors very promising and give them important advantages over the widely spread fiber optic sensors.

## 4. Conclusions

The concept of using polymer top-covered Bragg stacks and polymer/metal thin film structures for optical sensing of humidity was verified and confirmed. The sensing medium was a thin film of branched poly(*N*,*N*-dimethylacrylamide)-based copolymer with optimized thickness. The sensitivity and resolution of detection as well the dynamic range were compared to the case of single film on silicon substrate. The widest dynamic range of sensing (5–90% RH) was obtained for polymer with a thickness of 100 nm deposited on silicon substrate. The sensitivity was 0.13%, while the resolution was 2% RH. The disadvantage to this method is that, in this case, monitoring of reflectance should be applied. The optimal sensing configuration was a 300 nm thick polymer film deposited on a Au–Pd thin film with a thickness of 30 nm and measuring transmittance signal at two appropriately chosen wavelengths. A sensitivity of 0.14 % was achieved for the 25–70% RH range, which increased to 0.39% for higher humidity (70–90% RH). Relative humidity of 0.7% and 0.3% could be resolved, respectively. There was a narrowing of the dynamic range (25–90% RH), which is acceptable. When a top-covered Bragg stack was used as a humidity sensor a sensitivity of 0.37% and resolution of 0.3% RH could be achieved for a relative humidity range of 40–80% RH.

## Figures and Tables

**Figure 1 nanomaterials-09-00875-f001:**
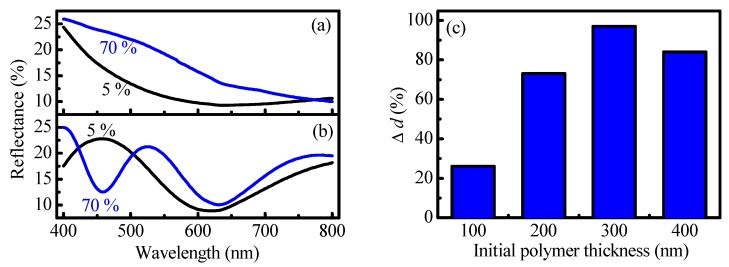
Left: Reflectance spectra at relative humidity (RH) of 5% and 70% of polymer films with initial thickness of 100 nm (**a**) and 300 nm (**b**) deposited on Si substrate; Right: Relative thickness change Δ*d* (in percent) of thin polymer films with initial thicknesses in the range 100–400 nm due to the change of humidity from 5% RH to 70% RH (**c**).

**Figure 2 nanomaterials-09-00875-f002:**
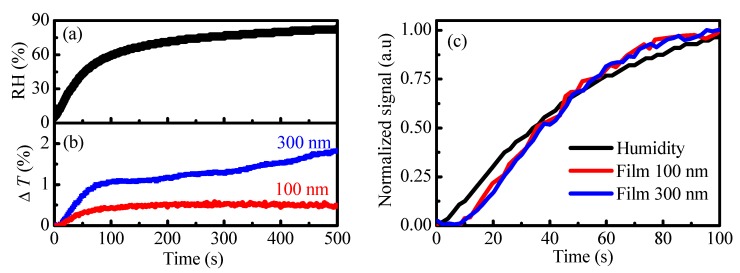
Left: (**a**) Temporal change of the humidity in the cell measured by the reference humidity sensor; (**b**) temporal change of transmittance of thin polymer films with denoted thicknesses deposited on glass substrates when humidity changes gradually from 5% RH to 80% RH; Right: (**c**) Normalized signal for temporal changes of humidity and transmittance of polymer films with thicknesses of 100 and 300 nm.

**Figure 3 nanomaterials-09-00875-f003:**
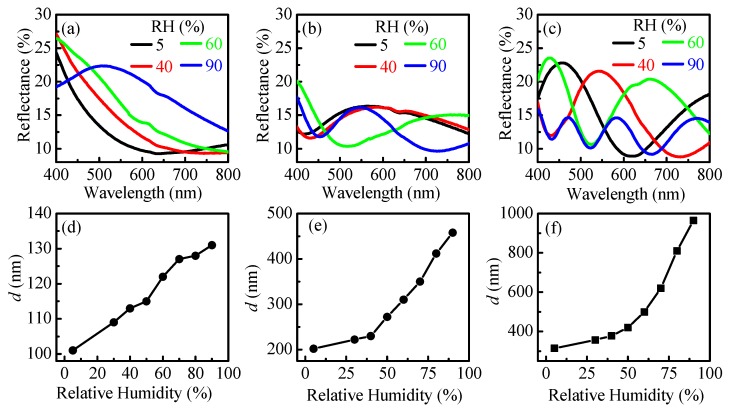
Reflectance spectra at denoted relative humidity values of thin polymer films with thickness of 100 nm (**a**), 200 nm (**b**), and 300 nm (**c**) deposited on silicon substrates. Growth of film thickness with relative humidity for thin polymer films with initial thicknesses of 100 nm (**d**), 200 nm (**e**), and 300 nm (**f**).

**Figure 4 nanomaterials-09-00875-f004:**
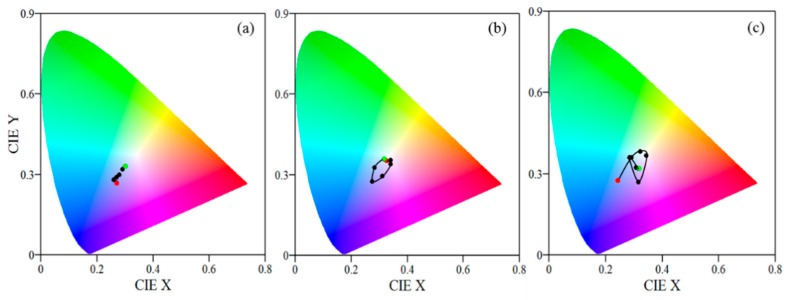
CIE color coordinates for polymer films with initial thickness of 100 nm (**a**), 200 nm (**b**), and 300 nm (**c**) deposited on silicon substrate and exposed to different humidity levels starting from 5% RH (red circle) and ending at 90% RH (green circle). The middle values are 30%, 40%, 50%, 60%, 70%, and 80% RH.

**Figure 5 nanomaterials-09-00875-f005:**
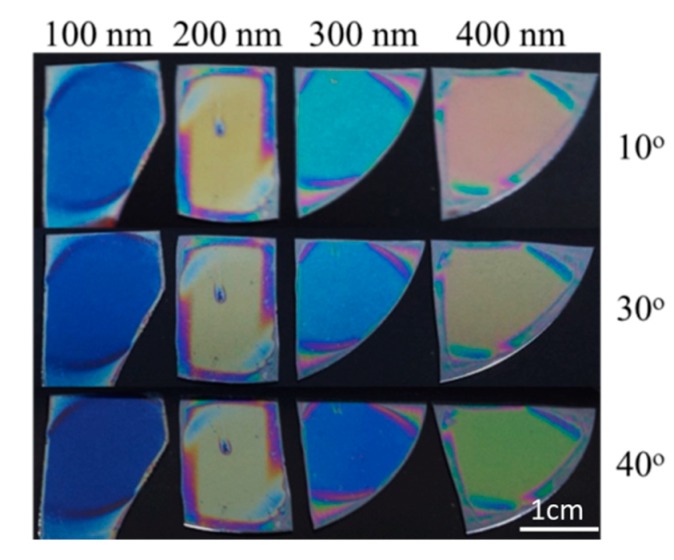
Pictures of polymer films with thicknesses of 100, 200, 300, and 400 nm deposited on silicon substrate at three viewing angles—10, 30, and 40 degrees.

**Figure 6 nanomaterials-09-00875-f006:**
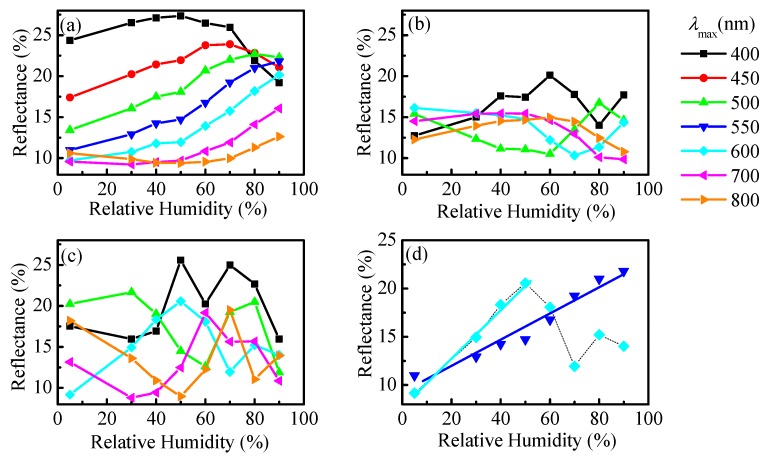
Reflectance at different wavelengths λ_max_ measured as a function of relative humidity for films with initial thicknesses of 100 nm (**a**), 200 nm (**b**), and 300 nm (**c**) deposited on silicon substrates. The best results were obtained for thin film with an initial thickness of 100 nm (blue triangles) and 300 nm (cyan diamonds) (**d**).

**Figure 7 nanomaterials-09-00875-f007:**
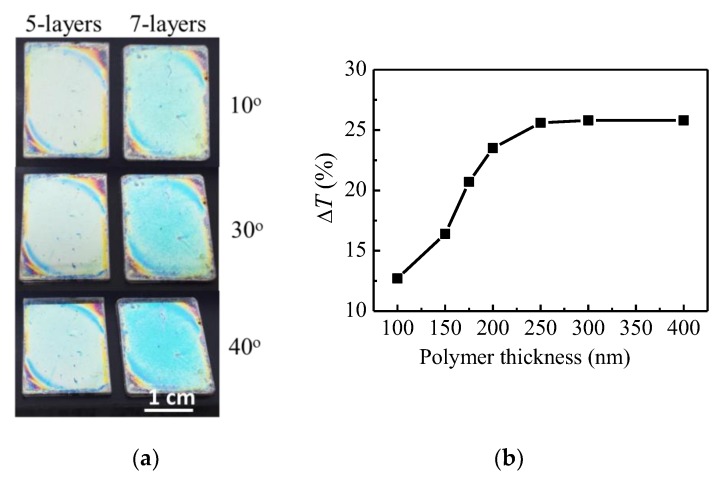
(**a**) Pictures of 5- and 7-layer Bragg stacks covered with polymer film with thicknesses of 300 at three viewing angles—10, 30, and 40 degrees; (**b**) calculated humidity-induced change of transmittance, Δ*T*, of 5-layer Bragg stacks covered with polymer film with different thicknesses.

**Figure 8 nanomaterials-09-00875-f008:**
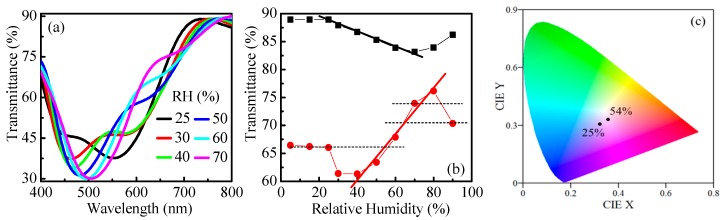
Transmittance spectra (**a**), transmittance values at λ_max_ of 650 nm (red circles) and 730 nm (black squares), (**b**) and calculated CIE color coordinates (**c**) of 5-layer Bragg stacks top-covered with 300 nm polymer film exposed to denoted levels of relative humidity.

**Figure 9 nanomaterials-09-00875-f009:**
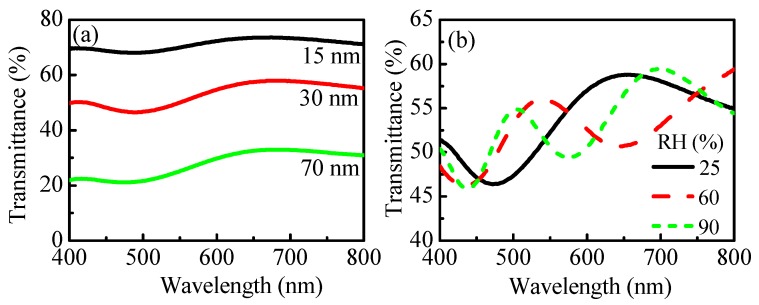
(**a**) Transmittance spectra of Au–Pd films with thicknesses of 15, 30, and 70 nm sputtered on glass substrates and covered with polymer film with a thickness of 300 nm. (**b**) Transmittance spectra of the system polymer (300 nm)/Au–Pd (30 nm)/glass at denoted humidity.

**Figure 10 nanomaterials-09-00875-f010:**
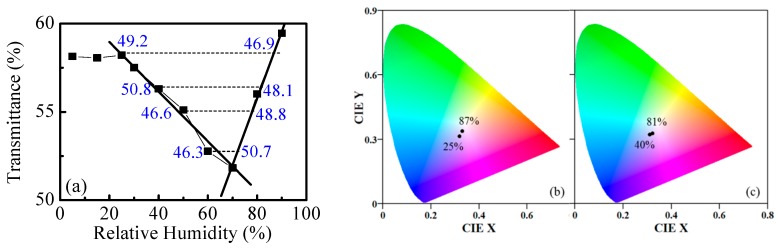
Humidity dependence of transmittance values at λ_max_ = 700 nm (**a**) and calculated CIE color coordinates (**b**,**c**) of polymer (300 nm)/Au–Pd film (30 nm)/glass exposed to denoted levels of relative humidity. The blue numbers denote transmittance values measured at a wavelength of 430 nm.

**Table 1 nanomaterials-09-00875-t001:** Dynamic range, RH, sensitivity, *S*, and resolution, Δ*RH*, calculated from Equations (1) and (2), respectively, measured signal type for detection, and sensitive medium.

RH (%)	*S* (%/% RH)	Δ*RH* (%)	Signal	Medium
5–90	0.13	2	*R* at 550 nm	100 nm/Si
5–50	0.26	1	*R* at 600 nm	300 nm/Si
40–80	0.37	0.3	*T* at 650 nm	300 nm polymer on Bragg stack
25–70	0.13	0.8	*T* at 730 nm	300 nm polymer on Bragg stack
25–90	0.14 (25–70) 0.39 (70–90)	0.7 (25–70) 0.3 (70–90)	*T* at 430 and 700 nm	300 nm polymer on Au/Pd (30 nm)

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
