# Peer review of "Optical Sensing of Humidity Using Polymer Top-Covered Bragg Stacks and Polymer/Metal Thin Film Structures"

_nanomaterials, 2019, doi:10.3390/nano9060875_

Reviewer 1 Report

The paper presents a detailed study on the humidity sensing by using polymer top-covered bragg stacks and polymer /metal thin film structures.  Although the manuscript studies most of the parameters and the performance of the deposited materials, the use of such architectures is not novel while the applicability is rather limited due to the restrictions in the interrogation approach.  Low cost and amplitude based fiber optic sensors functionalized with similar materials have been demonstrated, providing  a more robust and integrated solution.

However due to the excaustive analysis in a series of parameters in the performance of materials the manuscript might have a merit despite the fact  that the manuscript does not lie strictly within the "nanomaterials" area of interest  or in the focus of the Journal . 

To my view one strong issue that has not been examined in the reversibility of the humidity sensing and the corresponding time characteristics. This is a crucial issue to the applicability of materials in real applications. Authors should provide such study in some of the examples or at least and preferably for the top performing case.

Additionally I would suggest a drastic rewriting in some parts of the manuscript (Abstract, Introduction) as they seriously lack of clarity.

Author Response

We would like to thank to all reviewers for positive reading of our manuscript and for valuable comments and suggestions that they give us for improving the manuscript. I would like to assure the respected reviewers that all their comments and amendments are considered in the revised manuscript and the text is thoroughly edited. I hope that now our manuscript meets the quality criteria of the journal and will be considered for publication.

Sincerely,

Prof. Dr. Tsvetanka Babeva.

Reviewer 1

“…Although the manuscript studies most of the parameters and the performance of the deposited materials, the use of such architectures is not novel while the applicability is rather limited due to the restrictions in the interrogation approach.  Low cost and amplitude based fiber optic sensors functionalized with similar materials have been demonstrated, providing a more robust and integrated solution.”

Answer: We absolutely agree that Bragg stacks, consisting of different sensitive materials, have already been utilized as optical transducers of humidity changes. In our group we have developed such structures comprising films from nanosized zeolites as low refractive index materials and metal oxides or chalcogenide glasses as materials with high refractive index (see for example T. Babeva et. al.,  J. Mater. Chem., 22, 18136-18138, 2012; T. Babeva et. al., Dalton Trans., 43, 8868-8876, 2014; K. Lazarova et. al., Sensors, 14, 12207-12218, 2014). However, in this paper, instead of integrating the sensitive medium in a stack’s backbone, we deposited it on top of the stack, overcoming in this way the incompatibility issues that may arise, especially in the cases when polymers and inorganic materials are used as building blocks. To the best of our knowledge, such architectures have not been used yet. A possible reason is the presence of some ambiguity in detection when thicker films are used. As opposed to the conventional Bragg stack, the shift of the bang gap of top covered Bragg stack as a function of optical thickness is not so smooth (see Figure 8a). As a result the measured signal (transmittance or reflectance) at fixed wavelength as a function of optical thickness of sensitive medium is more complicated (see Figure 8b) and similar values of the signal are obtained at different humidity levels. In the manuscript we propose two approaches for overcoming this ambiguity – using simultaneous color detection and signal monitoring at two wavelengths.

“…To my view one strong issue that has not been examined is the reversibility of the humidity sensing and the corresponding time characteristics. This is a crucial issue to the applicability of materials in real applications. Authors should provide such study in some of the examples or at least and preferably for the top performing case”.

Answer: In our recent studies (Ref. 22 of the revised manuscript: K. Lazarova et. al., Polymers, 10 (7), 769, 2018) we have studied series of hydrophilic poly(N,N-dimethyl acrylamide)-based copolymers of different chemical composition and chain architecture such as triblock, star-shaped, and branched. We have demonstrated that macromolecular structure influences both the sensitivity and hysteresis level. It has been revealed that the most suitable polymer for optical sensing of humidity is the branched polymer, in which the optimal balance between sensitivity and hysteresis is achieved (Figure 7, ref. 22). Thin films with different thickness from this polymer have been used in the present study as sensitive medium and different transducing schemes have been proposed (single silicon substrate, Bragg stack and thin metal film on glass substrate). Considering that the reversibility of the humidity sensing depends on the intrinsic properties of the sensitive material and not on the transducing scheme, we have not presented the hysteresis results in the present manuscript because they have been already presented in ref. 22.

According to the time characteristics of the polymer thin films, in the present manuscript we have studied the response time of the sensor in the regime of continuous change of the humidity (Figure 2), because we think that this is more realistic case as compared to the case of stepwise increase of humidity. For relative humidity (RH) range 5 - 50 %, the humidity in the cell changes linearly with time at a rate of 0.8 %RH per second. The delay of our sensor with respect to the reference sensor is 8s at 10 %RH, 6s at 20 %RH and decreases to 4s for 30 %RH and 40 %RH. For rapid change of humidity from 50 %RH to 95%RH, achieved by blowing of humid air, the change of color is almost instant (see the recorded video). The video presents the change of color of polymer film with thickness of 300 nm deposited on silicon substrate during the blowing of humid air. It is seen that the sensor reaction is instant and it recovers rapidly when blowing is off.

 “Additionally I would suggest a drastic rewriting in some parts of the manuscript (Abstract, Introduction) as they seriously lack of clarity.”

Answer: The entire manuscript is thoroughly edited. A special attention is paid on the introduction and abstract parts as suggested by the reviewer. A lot of new references are added in the introduction as suggested by the third reviewer in order to clarify and strengthen further our points.

Reviewer 2 Report

The manuscript "Optical sensing of relative humidity using polymer top-covered Bragg filter stacks and polymer/metal thin film structures", written by Katerina Lazarova describes a development of advanced humidity sensor. The text is well written, and manuscript contains many new findings. However, I am missing a broader comparison of the sensor performance with existing technologies and deeper discussion of obtained results. The results are well summarized, however, discussion over their impact is limited to few words.

Author Response

We would like to thank to all reviewers for positive reading of our manuscript and for valuable comments and suggestions that they give us for improving the manuscript. I would like to assure the respected reviewers that all their comments and amendments are considered in the revised manuscript and the text is thoroughly edited. I hope that now our manuscript meets the quality criteria of the journal and will be considered for publication.

Sincerely,

Prof. Dr. Tsvetanka Babeva.

… I am missing a broader comparison of the sensor performance with existing technologies and deeper discussion of obtained results. The results are well summarized, however, discussion over their impact is limited to few words.”

Answer: The most widely spread and sensitive technology for humidity sensing is fiber optic sensing where the sensitivity is calculated as nm per %RH because for detection the wavelength shift of the signal dip is monitored (ref. 33 from the revised manuscript). From Table 1 of ref. 33 is seen that the sensitivity changes in broad range from 0.023nm/%RH to 1.01 nm / %RH. In order to make a comparison with our sensor, the resolution should be calculated (i.e the smallest RH step that could be distinguished, eq. 2 of the present manuscript). The wavelength accuracy, i.e. the measurement error, depends on the interrogation scheme used. If we assume value of 0.1 nm, the calculated resolution is 4.3 %RH for sensitivity of 0.023nm/%RH and 0.1%RH for 1.01 nm / %RH. Therefore, the value of 0.3 % achieved by our sensor is very close to the best reported resolution of optical humidity sensing so far. Furthermore, the proposed sensor has the advantage of simple preparation and simple detection. It comprises two planar thin films on glass substrate (sputtered thin Au-Pd film and spin-coated polymer film) and the detection is performed through two simple approaches – measuring transmittance signal at fixed wavelength or monitoring of color. The last could be performed even by the smartphone camera. The good sensitivity and technological convenience make the studied in this manuscript sensors very promising and give them important advantages over the widely spread fiber optic sensors. Unfortunately the last have disadvantages of relatively high cost of optical equipment, particularly spectrometers and optical spectrum analyzers, and complicated preparation of sensitive elements, especially in the case of side-polished optical fibers.

Reviewer 3 Report

Dear Editor

I accurately reviewed the manuscript Optical sensing of relative humidity using polymer top-covered Bragg stacks and polymer/metal thin film structures submitted to Nanomaterials.

The authors prepared thin films with nanometers thickness (100-400 nm) from double hydrophilic copolymer of complex branched structure containing poly(N,N-dimethyl acrylamide) and poly(ethylene oxide) blocks and these films were used as humidity sensitive media. Polymer thin films are deposited on two types of substrates - Bragg stacks and thin (30 nm) sputtered Au-Pd films thus bringing color for colorless polymer / glass system and enabling transmittance measurements for humidity sensing. All samples are characterized by transmittance measurements at different humidity levels in the range from 5 % to 90 % relative humidity. Additionally, the humidity induced color change is studied by calculating the color coordinates at different relative humidity. The influence of initial film thickness, sensor design and measuring configuration on sensitivity is studied.

The topic could be appropriate for this Journal. However, the authors must solve many critical issues.

Introduction

A more comprehensive introduction on polymeric nanomaterials, their preparation and applications as humidity sensors would be useful for the readers, citing literature, as examples:

Ultra-compact, fast-responsive and highly-sensitive humidity sensor based on a polymer micro-rod on the end-face of fiber core; Sensors and Actuators B: Chemical, Volume 290, 1 July 2019, Pages 23-27

Synthesis of conjugated polymeric nanobeads for photonic bandgap materials; Sensors and Actuators B 126 (2007) 35-40

Polymeric humidity sensors with nonlinear response: Properties and mechanism investigation; Applied Polymer Sci. 130,      2013, 2056-2061

Growth Control and Long range Self-assembly of Polymethylmethacrylate Nanospheres; Journal of Applied Polymer Sci 102(5), (2006) 4493-4499 

Smart Polymers in Micro and Nano Sensory Devices; Chemosensors 2018, 6(2), 12 

Gold nanoparticles in photonic crystals applications: a review; Materials 10(2), (2017) 97.

Humidity Sensors Principle, Mechanism, and Fabrication Technologies: A Comprehensive Review; Sensors (Basel). 2014 May; 14(5): 7881–7939.

From nanospheres to microribbons: Self-assembled Eosin Y doped PMMA nanoparticles as photonic crystals; J. Colloid Interf. Sci. 414 (2014) 24-32

An ingenious strategy for improving humidity sensing properties of multi-walled carbon nanotubes via poly-L-lysine modification; Sensors and Actuators B: Chemical, Volume 289, 2019, Pages 182-185

Self-assembled copolymeric nanoparticles as chemical interactive materials for humidity sensors; Nanotechnology 21 (2010) 355502 (8pp) 

Experimental

The materials used for nanoparticle synthesis must be reported with purity and supplier grade.

Details of the equipment used must be provided.

Details on the concentrations used for the characterizations must be provided.

details of the statistics used must be provided.

Experiments and measurements should be conducted at least in triplicate to obtain a mean and a standard deviation. This is to ensure the significance and reproducibility of the data.

Results

The part of the results is too fragmentary and does not discuss the preparation and deposition conditions.

Are figures 5 and figure 7a images from an optical microscope? Please, give details in experimental part

English needs many improvements: there are many typos and mistakes.

In conclusion, the paper could be suitable for publication in Nanomaterials, but after major revisions.

Best regards

Author Response

… A more comprehensive introduction on polymeric nanomaterials, their preparation and applications as humidity sensors would be useful for the readers…

Answer: Introduction part has been edited and all references suggested by the reviewer have been considered.

“The materials used for nanoparticle synthesis must be reported with purity and supplier grade

Details of the equipment used must be provided.

Details on the concentrations used for the characterizations must be provided.

Answer: All requested information is added in the revised version of the manuscript.

Details of the statistics used must be provided. Experiments and measurements should be conducted at least in triplicate to obtain a mean and a standard deviation. This is to ensure the significance and reproducibility of the data”

Answer: Indeed, the reflectance and transmittance spectra of the obtained thin polymer films were registered in at least three consecutive experiments for each film sample. As the reflectance and transmittance data points derived from the subsequent experiments were practically equal, standard deviations were not added to the figures in the manuscript. The reported accuracy of the spectrophotometric measurements and the calculated error in refractive index, extinction coefficient and thickness are  added in the revised version of the manuscript.

“The part of the results is too fragmentary and does not discuss the preparation and deposition conditions.”

Answer: The preparation and deposition conditions are discussed in part 2 of the manuscript “Materials and Methods”. The synthesis of copolymer and niobium sol is described in details in our previous papers [references 22 and 28 from the revised manuscript]. Besides, a brief description is also given in “Materials and Methods” part which was which has been supplemented and expanded according to all reviewers suggestions.

Are figures 5 and figure 7a images from an optical microscope? Please, give details in experimental part

Answer: Figures 5 and 7(a) are taken by camera, not by optical microscope. In the revised version of the manuscript a scale bar is added in order the dimension of the samples to be easily seen.

English needs many improvements: there are many typos and mistakes.

Answer: The entire manuscript is thoroughly edited.

Reviewer 4 Report

The article contains many inappropriate elements for a scientific article.

I exemplify only with the expression “Optical sensing of relative humidity” in title which denotes the lack of basic knowledge in the field of sensors. The “relative humidity” is a (relative) parameter, a humidity sensor measures humidity rather than “relative” humidity. After being measured, the humidity can be expressed relative, absolute etc.

Author Response

We would like to thank to all reviewers for positive reading of our manuscript and for valuable comments and suggestions that they give us for improving the manuscript. I would like to assure the respected reviewers that all their comments and amendments are considered in the revised manuscript and the text is thoroughly edited. I hope that now our manuscript meets the quality criteria of the journal and will be considered for publication.

Sincerely,

Prof. Dr. Tsvetanka Babeva.

The article contains many inappropriate elements for a scientific article. I exemplify only with the expression “Optical sensing of relative humidity” in title which denotes the lack of basic knowledge in the field of sensors. The “relative humidity” is a (relative) parameter, a humidity sensor measures humidity rather than “relative” humidity. After being measured, the humidity can be expressed relative, absolute etc.”

We are very grateful to the reviewer for having noticed this annoying mistake made in the title. We absolutely agree that the change of polymer film thickness is proportional to the humidity in its environment. In the revised version the term “relative” is removed from the title.

Round  2

Reviewer 1 Report

The revised version has considerably improved the clarity in several  points of the manuscript and I believe the paper has reached now a reasonable level for acceptance in Nanomaterials

Reviewer 3 Report

Dear Editor

I accurately reviewed the manuscript nanomaterials-511372-v2 Optical sensing of humidity using polymer top-covered Bragg stacks and polymer/metal thin film structures submitted to Nanomaterials.

The authors prepared thin films with nanometers thickness (100-400 nm) from double hydrophilic copolymer of complex branched structure containing poly(N,N-dimethyl acrylamide) and poly(ethylene oxide) blocks and these films were used as humidity sensitive media. Polymer thin films are deposited on two types of substrates - Bragg stacks and thin (30 nm) sputtered Au-Pd films thus bringing color for colorless polymer / glass system and enabling transmittance measurements for humidity sensing. All samples are characterized by transmittance measurements at different humidity levels in the range from 5 % to 90 % relative humidity. Additionally, the humidity induced color change is studied by calculating the color coordinates at different relative humidity. The influence of initial film thickness, sensor architecture and measuring configuration on sensitivity is studied.

In my opinion the topic is very interesting for readers of Nanomaterials and the authors solved the critical issues. The paper is now suitable for publication in Nanomaterials.

best regards

Reviewer 4 Report

-